# Simplified Point-of-Care Full SARS-CoV-2 Genome Sequencing Using Nanopore Technology

**DOI:** 10.3390/microorganisms9122598

**Published:** 2021-12-16

**Authors:** Anton Pembaur, Erwan Sallard, Patrick Philipp Weil, Jennifer Ortelt, Parviz Ahmad-Nejad, Jan Postberg

**Affiliations:** 1Clinical Molecular Genetics and Epigenetics, Centre for Biomedical Education & Research (ZBAF), Faculty of Health, Witten/Herdecke University, Alfred-Herrhausen-Str. 50, 58448 Witten, Germany; anton.pembaur@uni-wh.de (A.P.); patrick.weil@uni-wh.de (P.P.W.); 2Institute of Virology and Microbiology, Centre for Biomedical Education & Research (ZBAF), Faculty of Health, Witten/Herdecke University, Stockumer Str. 10, 58453 Witten, Germany; erwan.sallard@uni-wh.de; 3Institute of Medical Laboratory Diagnostics, Centre for Clinical & Translational Research (CCTR), HELIOS University Hospital Wuppertal, Witten/Herdecke University, Heusnerstr. 40, 42283 Wuppertal, Germany; jennifer.ortelt@helios-gesundheit.de (J.O.); parviz.ahmad-nejad@helios-gesundheit.de (P.A.-N.)

**Keywords:** (+)RNA genome sequencing, COVID-19 surveillance, variant-of-concern (VOC), variant-of-interest (VOI)

## Abstract

The scale of the ongoing SARS-CoV-2 pandemic warrants the urgent establishment of a global decentralized surveillance system to recognize local outbreaks and the emergence of novel variants of concern. Among available deep-sequencing technologies, nanopore-sequencing could be an important cornerstone, as it is mobile, scalable, and cost-effective. Therefore, streamlined nanopore-sequencing protocols need to be developed and optimized for SARS-CoV-2 variants identification. We adapted and simplified existing workflows using the ‘midnight’ 1200 bp amplicon split primer sets for PCR, which produce tiled overlapping amplicons covering almost the entire SARS-CoV-2 genome. Subsequently, we applied Oxford Nanopore Rapid Barcoding and the portable MinION Mk1C sequencer combined with the interARTIC bioinformatics pipeline. We tested a simplified and less time-consuming workflow using SARS-CoV-2-positive specimens from clinical routine and identified the CT value as a useful pre-analytical parameter, which may help to decrease sequencing failures rates. Complete pipeline duration was approx. 7 h for one specimen and approx. 11 h for 12 multiplexed barcoded specimens. The adapted protocol contains fewer processing steps and can be completely conducted within one working day. Diagnostic CT values deduced from qPCR standardization experiments can act as principal criteria for specimen selection. As a guideline, SARS-CoV-2 genome copy numbers lower than 4 × 10^6^ were associated with a coverage threshold below 20-fold and incompletely assembled SARS-CoV-2 genomes. Thus, based on the described thermocycler/chemistry combination, we recommend CT values of ~26 or lower to achieve full and high-quality SARS-CoV-2 (+)RNA genome coverage.

## 1. Background

To face the ongoing SARS-CoV-2 (Severe Acute Respiratory Syndrome Coronavirus 2) pandemic, a global decentralized warning system is being established to recognize local outbreaks and the emergence of novel variants of concern (VOC). A particular focus is given to the identification of VOCs with accelerated transmission rates, increased infectivity or immune escape mutations as these variants would warrant adaptations in containment and vaccination strategies [1,2,3,4]. Moreover, the search for the zoonotic origin of SARS-CoV-2 from comparative analyses of genomic data is an ongoing issue with relevance for the early recognition of future outbreak scenarios [5,6].

Among the available deep-sequencing technologies, nanopore-sequencing could be an important cornerstone, since it is mobile, scalable and acquisition investments are comparatively low. Further, nanopore sequencing devices do not require large-scale information technology (IT) infrastructure. They were already involved in genome surveillance, e.g., during the 2015 Ebola outbreak in Liberia, Guinea and Sierra Leone [7], even though nanopore technology was not as developed as today. However, at least for smaller hospital laboratories with lower throughput, it is still desirable to develop further streamlined protocols.

Nanopore sequencing allows the sequencing of either DNA or RNA [8], and does not require PCR amplification. Furthermore, the technique has the potential of producing very long, continuous reads, which theoretically allows to sequence in only one read the 29.903 nt long (+)RNA genome of SARS-CoV-2 [9], or its deriving cDNA after reverse transcription (Figure 1).

With the aims of facilitating implementation in routine diagnostics with lower specimens throughput and of simplifying the workflow, we tested modifications of existing ARTIC protocols for SARS-CoV-2 full length (+)RNA genome sequencing [11,12,13]. Furthermore, we tested the simplified and less time-consuming workflow on confirmed SARS-CoV-2-positive specimens from clinical routine and identified parameters, most importantly CT values corresponding to standardized viral genome copies, which may help to decrease the rate of sequencing failures.

## 2. Methods

### 2.1. Pre-Analytics

Specimens used in this study included mainly nasopharyngeal swabs (Xebios Diagnostics) routinely sampled from non-hospitalized individuals by public health authorities. Before study inclusion as anonymized specimens, routine COVID-19 diagnostic testing was conducted. 

### 2.2. Nucleic Acids Purification

Total RNA was extracted from 250 µL liquid specimen using either 750 µL QIAzol lysis reagent (Qiagen, Hilden, Germany) upon manufacturer’s recommendations or purification via magnetic beads (Seegene NIMBUS/Tanbead, Seoul, Korea) or silica columns (QiaAmp Viral Mini Kit, Qiagen). 

### 2.3. RT-PCR, Quality Assessments and Library Preparation

The ‘midnight’ split primer set from the ARTIC protocol was used for SARS-CoV-2 cDNA amplification in 2 multiplex PCR reactions [11]. To avoid overlaps during multiplex PCR, each single-tube PCR reaction generates consecutively tiled, non-overlapping 1200 bp amplicons. Mixed together after PCR, both resulting complementary amplicon mixtures cover almost the entire SARS-CoV-2 genome. 

Combined reverse transcription and amplification of multiple 1200 bp amplicons (RT-PCR) were performed in single tube 20 µL reactions using the Luna One-Step RT-qPCR Kit (NEB; E3005). For RT-PCR, 8 µL of purified template RNA were used for each reaction. 1 µL of 100 µM primer pool was used in each reaction. Reverse transcription was performed at 55 °C for 30 min, followed by incubation at 95 °C for one minute. Then 34 cycles (pool 1) or 30 cycles (pool 2) of denaturation at 95 °C for 20 s and annealing and extension in one step at 60 °C for 210 s were performed. A final extension was performed at 65 °C. During the implementation phase, amplicon sizes and DNA concentrations were routinely checked by agarose gels or by microvolume electrophoresis using an Agilent Bioanalyzer instrument and a microfluidic chip (Agilent DNA 12,000 kit, Agilent). Thereafter, amplicons from primer pools 1 and 2 were quantified using the QuantiFluor dsDNA System (Promega, Madison, WI, USA) with the Promega Quantus fluorometer and then mixed at equal concentrations. Library preparation was done using the Rapid Barcoding Sequencing Kit (SQK-RBK004; Oxford Nanopore Technologies, Oxford, UK) upon manufacturer’s recommendations.

### 2.4. Nanopore Sequencing

Sequencing was performed on a MinION Mk1C (Oxford Nanopore Technologies) with the options ‘basecalling’ and ‘demultiplexing’ being enabled, both performed by the included ‘guppy’ algorithm. As output format, FAST5 and FASTQ files were chosen. Sequencing time was set for 72 h as default. Sequencing was stopped after reaching at least 10 megabases for each barcode.

### 2.5. Bioinformatics

Consensus sequences were built from the barcode-sorted, quality-filtered FAST5 and FASTQ files containing sequencing reads, using the interARTIC pipeline. Except otherwise stated, the ‘Nanopolish’ algorithm was routinely used.

Installation and usage of interARTIC pipeline were done following the developers’ instructions: InterARTIC Documentation. Available online: https://psy-fer.github.io/interARTIC/installation/ (accessed on 14 December 2021). For faster analysis, all available threads were activated in the advanced settings. The interARTIC pipeline performs read filtering, alignments and returns a consensus FASTA file as well as coverage charts for visualization (Figure 2B–H).

Consensus FASTA files were uploaded to the Nextstrain webapp (Nextclade. Available online: https://clades.nextstrain.org [accessed on 14 December 2021]) to perform phylogenetic analyses [14].

### 2.6. Data Availability

FASTQ files and assembled FASTA formatted consensus sequences are available (Genome Sequence Archive (GSA). Available online: https://bigd.big.ac.cn/gsa/browse/CRA005542 [accessed on 14 December 2021]).

## 3. Results and Discussion

We defined milestones to simplify the protocol and decrease hands-on sample time and working step numbers. 1. We tested whether specimens can be directly taken from residual diagnostic specimens extracted from 96-deepwell-plates using magnetic beads (Seegene NIMBUS/Tanbead). For comparison, we applied RNA purification protocols using silica columns (QiaAmp Viral Mini Kit, Qiagen) or guanidinium isothiocyanate (GITC) for RNA extraction (QIAzol lysis reagent, Qiagen). 2. We tested whether reverse transcription can be successfully primed by SARS-CoV-2-specific primers, which are subsequently used for multiplex 1200 bp amplicon amplification. Purified RNA from the different extraction protocols of milestone 1 was used for cDNA synthesis and successive multiplex PCR with the 2 ‘midnight’ split 1200 bp amplicon primer pools in single tube reactions. 3. We aimed at improving the efficiency of the nanopore sequencing workflow by using the onboard Guppy basecalling capability of the Oxford Nanopore MinION Mk1C device. Moreover, implementation of the interARTIC interface should help to avoid command line-based bioinformatics analyses as much as possible to provide a user-friendly and efficient analysis pipeline.

### 3.1. Protocol Implementation Considering Differences in Viral Loads and Influences of RNA Extraction Protocols

Sequencing was performed from serially diluted specimens of purified RNA from patients’ samples, which exhibited low cycle threshold (CT) value (CT = 16–18) after routine diagnostics RT-qPCR (N gene, RdRP gene; Seegene). Using the quantitative reference sample Ch07470 for calibration we determined that a CT = 25 (respectively 16 and 18) corresponded to a SARS-CoV-2 copy number of 1.0 × 10^6^ (respectively 5.12 × 10^8^ and 1.28 × 10^8^). Routinely used dilution factors were 2^0^–2^−10^ to cover several CT value magnitudes and to simulate different amounts of viral loads. In terms of RNA yield and 1200 bp amplicon PCR performance, the magnetic beads-based RNA purification protocol outperformed slightly the GITC method as well as the column-based protocol, but we did not observe differences in read and coverage quality between these different isolation methods (Appendix A). Since direct sampling from 96-deepwell plates allowed us to directly exploit residual specimens, which remained after routine RT-qPCR diagnostics, we decided to focus on magnetic beads-based RNA purification during further protocol development. Moreover, it shortened and simplified the workflow.

After semiquantitative or quantitative RT-PCR using multiplex primer pools 1 and 2 in separate single-tube reactions for combined reverse transcription of the SARS-CoV-2 (+)RNA and amplification of 1200 bp amplicons, band intensities exhibited a strong dependence on viral loads. Moreover, after 32 PCR cycles band intensities using primer pool 1 were weaker when compared with primer pool 2 (Appendix A). We determined that 34 PCR cycles for primer pool 1 and 30 cycles for primer pool 2 were a good compromise. Therefore, reverse transcription can be successfully primed by the ‘midnight’ primers.

### 3.2. Surveillance of Multiplex Nanopore Sequencing and Multiple Reuses of Flow Cells

Using the Rapid Barcoding Kit (SQK-RBK004, Oxford Nanopore) enabled us to barcode the PCR products without purification steps and sequence them immediately. As a standard, we used 12 barcoded libraries for multiplex sequencing on R9 flow cells. In contrast to the Oxford Nanopore MinION Mk1B, the MinION Mk1C device features onboard guppy basecalling. In combination with the Rapid Barcoding Kit, we exploited this opportunity for real-time surveillance of basecalling and demultiplexing for each of the 12 multiplexed samples per run. This enabled us to recognize the exact time point at which a reading depth of approx. 10 Mbp per barcode was achieved. The time to reach this threshold depended heavily on the viral load (simulated by serially diluted samples). Decreased viral loads led to a considerable decrease of passed reads (Figure 2A). A manual stop of sequencing followed by flow cell washing (Flow Cell Wash Kit, EXP-WSH004, Oxford Nanopore) allowed us to reuse a single flow cell for a series of 3 sequencing runs, each using 12 multiplexed barcoded libraries. This specific scenario resulted in costs of approx. 40 USD per sample. Theoretically, the possible number of reuses depends largely on the duration of the sequencing, which itself depends mainly on the number of used barcodes. Further, if a MinION Mk1B is considered for this purpose, a similar surveillance functionality could be achieved using RAMPART (artic-network/rampart. Available online: https://github.com/artic-network/rampart [accessed on 14 December 2021]) on a dedicated LINUX environment. RAMPART runs simultaneously with MinKNOW and exhibits demuxing and mapping results in real-time.

### 3.3. Assembly of Full SARS-CoV-2 Genomes and Pathogen Genome Data Analyses

For mapping and full-length SARS-CoV-2 genome assembly, we used the FASTQ files resulting from Guppy basecalling to generate FASTA formatted consensus sequence files. We used the ARTIC pipeline through a graphical user interface (Psy-Fer/interARTIC. Available online: https://github.com/Psy-Fer/interARTIC [accessed on 14 December 2021]). Once installed, this is an easy to use and relatively fast pipeline with only five minutes of hands-on time, which enables the use of the ‘Nanopolish’ or ‘Medaka’ algorithms for simultaneous analyses of multiplexed barcoded samples. As an example, Figure 2B shows the complete and deep coverage of the complete SARS-CoV-2 genome after the combination of pools 1 (light blue) and 2 (pink). This demonstrates that all contained 1200 bp amplicon were specifically and efficiently amplified during the combined RT-PCR reaction (Figure 2B).

To compare the interARTIC and Geneious Prime pipelines, we used exactly the same FASTQ file from the same sample shown in Figure 2B (geneious—How to assemblecoronavirus genomes. Available online: https://go.geneious.com/video/how-to-assemble-coronavirus-genomes [accessed on 14 December 2021]). Phylogenetic analyses with Nextstrain using the FASTA consensus files obtained from the interARTIC or the Geneious Prime pipelines resulted in considerably different phylogenetic distances in clade 20I, showing that the bioinformatics pipeline influences the result (Appendix A). Notably, despite the calculated different phylogenetic distances using Geneious, the sample was still assigned to clade 20I. We speculate that the reason for this distance bias might be due to the Illumina-optimized assembly pipeline of Geneious. The interARTIC pipeline proved superior in terms of coverage and sequencing depth. Notably, within the interARTIC pipeline both options, the ‘Nanopolish’ and ‘Medaka’ algorithms performed equally well with respect to consensus sequence quality, but ‘Medaka’ was considerably faster.

We observed that the consensus sequences returned by the ‘Nanopolish’ and ‘Medaka’ algorithms contain numerous unsolved regions (for which only ‘N’s are indicated). Interestingly, the regions unsolved by one algorithm were generally solved by the other, so we assumed that the two consensus sequences could be combined to produce a ‘super-consensus’ with improved variant prediction value. We developed a Python code that merges the ‘Nanopolish’ and ‘Medaka’ consensus sequences and generates the corresponding variant calling file. As expected, the ‘super-consensus’ contained fewer unsolved regions when compared with the ‘Nanopolish’ and ‘Medaka’ consensus sequences alone (Figure 3B). In addition, it retained the high-quality mutations, which were identified by both algorithms, while removing most false positives probably caused by sequencing and alignment errors (Figure 3A). Consequently, our consensus-merging code improves the quality of variant calling and highlights the complementarity of ‘Nanopolish’ and ‘Medaka’ for nanopore-sequencing of SARS-CoV-2 and others. Nevertheless, this quality increase comes at the cost of information loss such as the number of reads per variant or other metadata which were initially generated by ‘Nanopolish’ and ‘Medaka’ and are not transferred to the ‘super-consensus’.

Serial input RNA dilutions or, respectively, viral load influenced the depth of sequencing (Figure 2C–H). For the output of high-quality consensus sequences in the FASTA file format, a coverage threshold of 20 was used as default. We generally observed that this could be reached when a SARS-CoV-2 titer of 4 × 10^6^ was given. Viral copy numbers lower than 4 × 10^6^ were associated with incompletely assembled SARS-CoV-2 genomes. Thus, we provide here a convincing line of evidence that the copy number-normalized CT values of diagnostic RT-qPCR can be used as the criterion of sequencing success.

Single or batch high-quality consensus FASTA formatted sequences were used for phylogenetic tree visualization and variant calling using the Nextstrain webapp (Nextclade. Available online: https://clades.nextstrain.org [accessed on 14 December 2021]) [14]. In our hands, the obtained sequences could faithfully be assigned to specific clades in the reference tree (Figure 4). Again, an influence of viral load was observed. However, despite incomplete coverage in those cases enough informative sequence data could be obtained for phylogenetic analyses from several low copy number samples. As a result of serial sample dilutions, we observed deviating phylogenetic distances within the clade, wherein specimens classification occurred (Figure 4A), which eventually could lead to incorrect clade association. The use of specimens from diagnostic routine with viral copy numbers higher than approx. 4 × 10^6^ apparently led to their faithful association with different clades, which were clades 20I and 19A in the shown example (Figure 4B). The ‘Nanopolish’ and ‘Medaka’ consensus as well as the merged super-consensus could be reliably associated with the corresponding clade (20I in the shown example). The ‘Medaka’ consensus mapped at a greater distance than the ‘Nanopolish’ consensus, probably because the ‘Medaka’ algorithm does not correct frameshifts, while the super-consensus had an intermediary distance between the two other consensuses (Figure 3C).

Taken together, the main achievements of an optimized workflow are 1. Purified RNA from SARS-CoV-2-positive patients can be directly taken from residual diagnostic specimens in 96-deepwell-plates; 2. cDNA synthesis and successive multiplex PCR with 2 split primer pools can be performed in single tube reactions. Since cDNA synthesis is primed by SARS-CoV-2-specific primers for 1200 bp amplicon amplification, there is no need for use of unspecific hexanucleotide priming, which are unspecific for SARS-CoV-2 (+)RNA during cDNA synthesis and could thus influence subsequent PCR reactions; 3. Onboard Guppy basecalling with the Oxford Nanopore MinION Mk1C device and implementation of the interARTIC led to a further reduction of working steps and hands-on time (Figure 5). Implementation in smaller hospital laboratories with lower specimens’ throughput can be easily done at moderate costs.

We provide a detailed protocol for all steps here, which includes the Python code (Appendix A) and corresponding command line: nanopore nCoV-2019 sequencing protocol. Available online: dx.doi.org/10.17504/protocols.io.bx72prqe (accessed on 12 December 2021).

## 4. Conclusions

The adapted protocol contains fewer processing steps than previous workflows (Figure 5). Diagnostic CT values are the principal criteria for specimen selection as variants could be accurately identified for initial viral loads of 4 × 10^6^ or above. After diagnostic qRT-PCR, multiplex library preparation, quality controls, nanopore sequencing and the bioinformatics pipeline can be completely conducted within one working day.

## Figures and Tables

**Figure 1 microorganisms-09-02598-f001:**
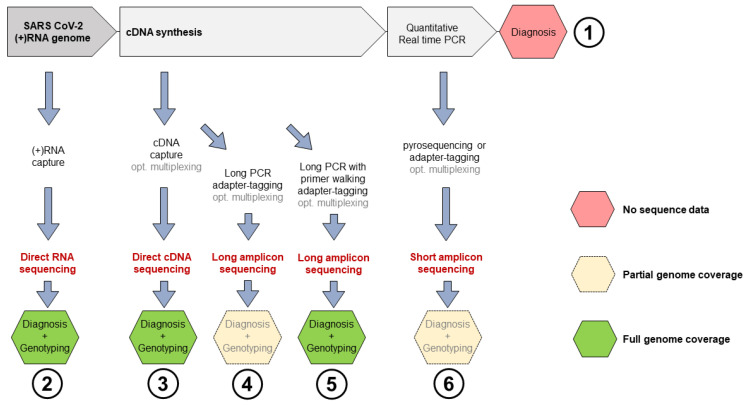
Nanopore sequencing options (**2**–**6**) compatible with diagnostic qPCR pipelines (**1**). The pyrosequencing option in the path (**6**) requires biotin-tagged primers for qPCR [10].

**Figure 2 microorganisms-09-02598-f002:**
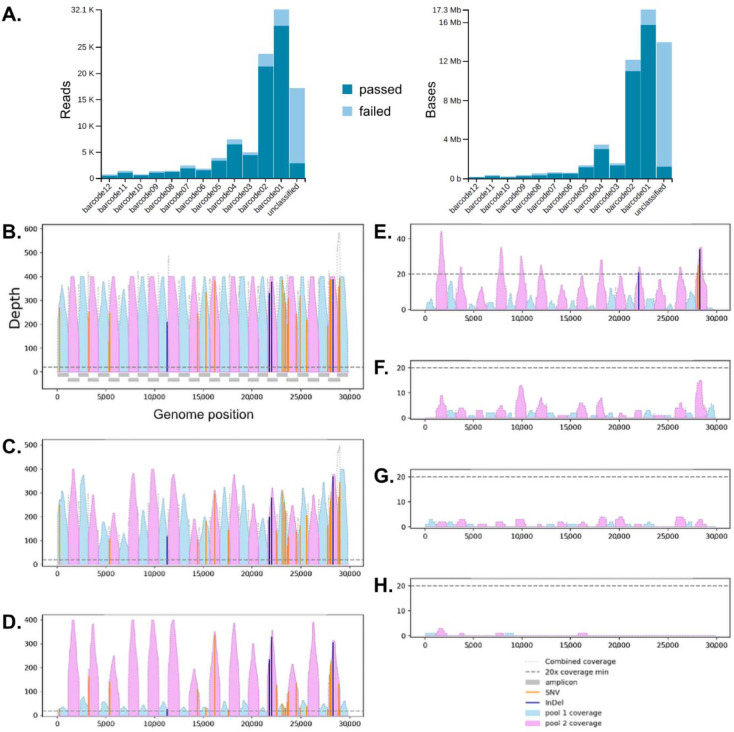
Surveillance using nanopore sequencing (**A**), and effects of specimens dilution on the SARS-CoV-2 genome coverage (**B**), whereby viral copy numbers were 1.28 × 10^6^ (**B**); 5.12 × 10^8^ (**C**); 2.56 × 10^8^ (**D**); 4 × 10^6^ (**E**); 2 × 10^6^ (**F**); 1 × 10^6^ (**G**); 5 × 10^5^ (**H**). In (**A**) increasing barcode numbers (*X*-axis) correspond to decreasing viral titers.

**Figure 3 microorganisms-09-02598-f003:**
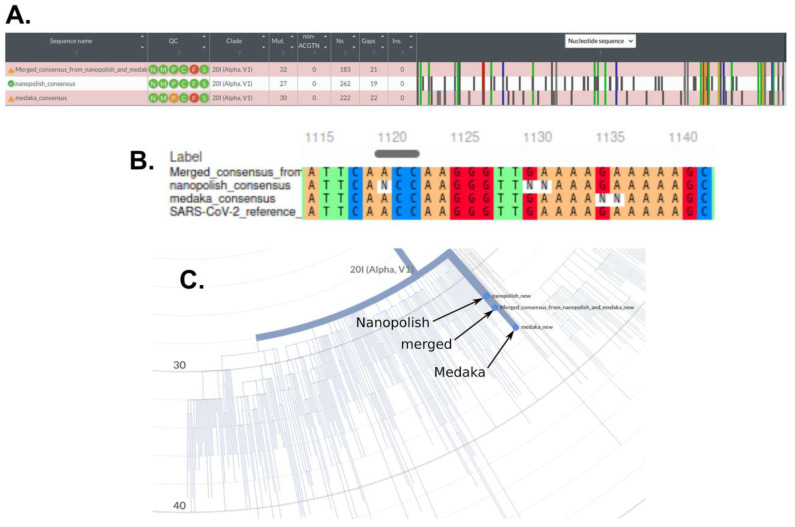
Comparison of the consensus sequences returned by ‘Nanopolish’ and ‘Medaka’ algorithm and the merged consensus for a nanopore sequencing run corresponding to a viral load of 1.28 × 10^6^. (**A**) Alignment of the ‘Nanopolish’ and ‘Medaka’ consensus as well as the merged consensus on the SARS-CoV-2 reference genome by the Nextstrain program. The merged consensus conserved all of the high-quality mutations that mapped to the known variant 20I (shown in color or light grey), while most non-matching mutations (in dark grey, likely sequencing errors) of the ‘Nanopolish’ or ‘Medaka’ consensus were lost; (**B**) Comparison of variant solving by ‘Nanopolish’, ‘Medaka’ and our code. The multiple alignments were performed by MAFFT online tool with the three consensus sequences and the SARS-CoV-2 reference genome (MN908947.3). Contrary to the ‘Nanopolish’ and ‘Medaka’ consensus, the merged consensus solved the entire region and led to accurate variant calling; (**C**) Clade mapping and phylogenetic distances calculated by Nextstrain for the three consensus sequences.

**Figure 4 microorganisms-09-02598-f004:**
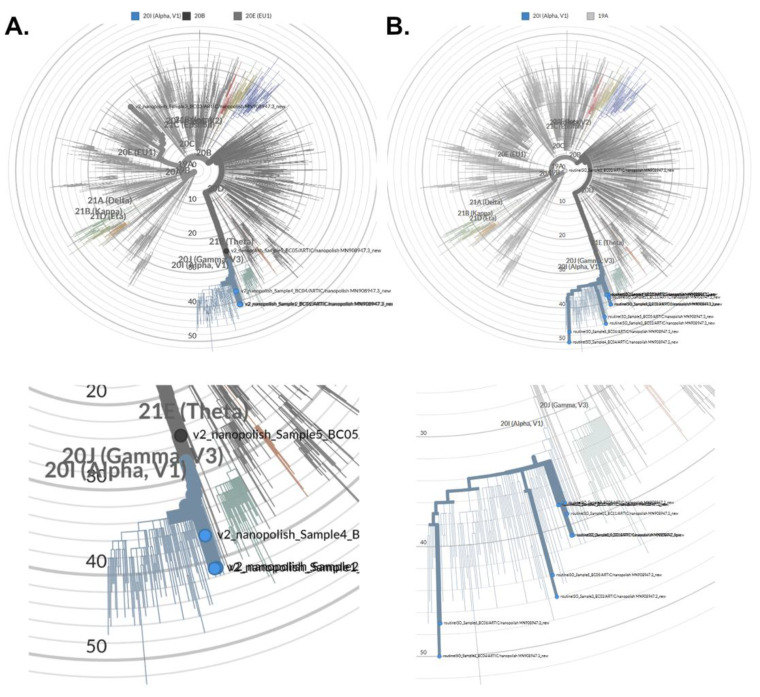
Comparison on the evolutionary distance calculation, when data from the same specimen was processed by 2 different bioinformatics pipelines. Phylogenetic tree visualization was done using the Nextstrain open-source platform for pathogen genome data analyses [14]. We used serial dilutions of identical specimens (**A**), or a selection of samples from different individuals from clinical routine (**B**) for phylogenetic analyses. Below the trees, a magnification of clade 20I is shown, wherein most specimens are grouped (**A**,**B**).

**Figure 5 microorganisms-09-02598-f005:**
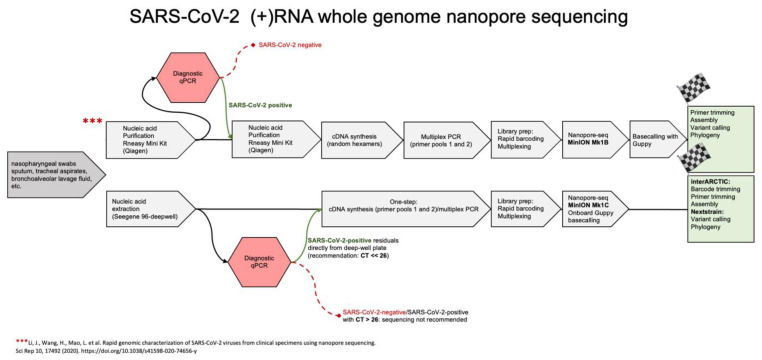
Comparison of a rapid SARS-CoV-2 whole (+)RNA genome nanopore sequencing pipeline [11] with the novel simplified workflow, whose main achievements are: 1. Purified RNA from SARS-CoV-2-positive patients can be directly taken from residual diagnostic specimens in 96-deepwell-plates; 2. cDNA synthesis and successive multiplex PCR with 2 primer pools can be performed in single tube reactions. Since cDNA synthesis is primed by SARS-CoV-2-specific primers for 1200 bp amplicon amplification, there is no need for use of unspecific hexanucleotide priming; 3. Onboard Guppy basecalling with the Oxford Nanopore MinION Mk1C device and implementation of the interARTIC led to a further reduction of working steps and hands-on time.

## Data Availability

FASTQ files and assembled FASTA formatted consensus sequences are available (Genome Sequence Archive (GSA). Available online: https://bigd.big.ac.cn/gsa/browse/CRA005542 [accessed on 14 December 2021]).

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
