# Peer review of "Simplified Point-of-Care Full SARS-CoV-2 Genome Sequencing Using Nanopore Technology"

_microorganisms, 2021, doi:10.3390/microorganisms9122598_

Round 1

Reviewer 1 Report

Pembaur and colleagues have adapted an existing workflow based on the well-known midnight primer set to the application of ONP  for SARS-CoV-2 detection and analysis in a reasonable amount of time, which would be of particular importance for running such analysis locally.

The paper is well written an all the required details can be found within the main text or in the supplementary materials, including the steps for data analysis in a reasonable detail. I noticed that the deposited data are not currently available, and the quality of some figures is low.

In conclusion, I recommended the publication of the paper after minor revisions detailed below.

Background:

Please, spell the name of SARS-CoV-2 at the first use in the Background section

Material section:

If possible, you can anticipate or repeat the reference for the “midnight primer set” after mentioning it and not at the end of the following sentence.

Mention first the kit name and then the manufacturer (e.g. for Agilent DNA kit)

An important point is that the quality of Figure 4 is low and this impairs its correct understanding.  Similarly, the tree labels of figure 3 are unreadable. Figure 3 arrives after figure 4, please correct figure ordering or numbering. Track changes are present in the legend of figure 3.

Conclusions should be detailed a bit more, e.g., the new protocol contains less steps of what?

Author Response

Dear reviewer 1,

we appreciate very much your fast handling and constructive criticism on our manuscript. Please find below our point-by-point responses to your queries, which surely helped to improve our revised manuscript.

I noticed that the deposited data are not currently available, and the quality of some figures is low.

The data at https://t1p.de/minion-seqdata is indeed unavailable (the link opens a “404 not found” page), although I remember that it was in July.

Response: Thank you for this important issue. We have moved the raw data to another genome data repository (Genome Sequence Archive [GSA]), which hopefully will be stable. Moreover, we have replaced low quality figures by figures with improved resolution.

Background:

Please, spell the name of SARS-CoV-2 at the first use in the Background section.

Response: We have made this change.

Material section:

If possible, you can anticipate or repeat the reference for the “midnight primer set” after mentioning it and not at the end of the following sentence.

Response: We have repositioned the citation as suggested.

Mention first the kit name and then the manufacturer (e.g. for Agilent DNA kit)

Response: We have made these changes.

An important point is that the quality of Figure 4 is low and this impairs its correct understanding.  Similarly, the tree labels of figure 3 are unreadable. Figure 3 arrives after figure 4, please correct figure ordering or numbering. Track changes are present in the legend of figure 3.

Response: We have corrected the figures' order and replaced low quality figures by figures with improved resolution.

Conclusions should be detailed a bit more, e.g., the new protocol contains less steps of what?

Response: Figure 5 and Legend to Figure 5 illustrate/describe in detail where our simplified workflow contains less working steps. To avoid redundancies, we decided to keep the manuscript text as it is. However, we propose to move Figure 5 to the Conclusions paragraph in the process of copy-editing.

Reviewer 2 Report

The authors present a timely and important work on the use of nanopore sequencing technology in hospital laboratories by facilitating and speeding the process for staff members who are not specialists in genomics. 

I only have a few minor comments. Otherwise, the paper is ready for publication on my end. 

Line 28: Maybe the authors could cite an example of the pre-analytical parameters they identified.

Line 32: I suggest adding the threshold Ct value for sequencing in the abstract.

Line 46-47: The acquisition investments are comparably low to other technologies? Please clarify

Line 47: Please consider defining the acronym IT.

Line 48-50: This sentence is hard to read - please consider re-writing and correcting grammatical errors.

Line 71-73: The connexion between the two sentences is unclear, and the use of the word alternatively is confusing.

Line 94: What ‘Basecalling’ tool was used? Any option choices?

Figure 3 is presented before figure 4

Line 117: missing comma

Line 124-126: The third objective is a bit confusing. Isn’t the Guppy algorithm already in-use by the nanopore software? What was the intent of the improvement?

Line 168-170: Can the authors develop more on this statement? How could a similar surveillance functionality be achieved using RAMPART?

Line 190-191: This is quite concerning. Could the authors develop more on the reasons behind these differences in the phylogenetic analysis based on the bioinformatic software used?

Line 227-229: The authors should provide a Ct value threshold for obtaining high-quality consensus sequences. This should also be provided in the abstract as it’s essential information for potential users of the proposed workflow.

Line 259: The unnecessity to use unspecific hexanucleotide priming is interesting, which is why the authors should develop more on this to give the readers a context of the use 

Author Response

Dear reviewer 2,

we appreciate very much your fast handling and constructive criticism on our manuscript. Please find below our point-by-point responses to your queries, which surely helped to improve our revised manuscript.

Line 28: Maybe the authors could cite an example of the pre-analytical parameters they identified.

Response: We have complemented the requested information in more detail in lines 28-39 (Abstract).

Line 32: I suggest adding the threshold Ct value for sequencing in the abstract.

Response: We expect that between labs, CT values differ depending on chemistry/thermocycler/threshold specifications. Therefore, in the manuscript text we avoid as much as we can to specify the exact CT but say genome coverage is complete with >=4x 10exp6 copies. We understand, however, that mentioning a CT value will be expected by many readers. To meet those expectations, we have now added a CT of approx. 26 as a rule-of-thumb threshold (see response above).

Line 46-47: The acquisition investments are comparably low to other technologies? Please clarify

Response: We avoid mentioning exact acquisition costs, since prizes and currencies change between different regions/countries and potentially over time. Moreover, regarding consumables several factors heavily influence costs per sample (e.g. degree of multiplexing, type of flow cells, multiple use of a flow cell). We believe that these information will be easy to acquire by potential users. Thus we do not list costs in the manuscript.

Line 47: Please consider defining the acronym IT.

Response: Change made as proposed.

Line 48-50: This sentence is hard to read - please consider re-writing and correcting grammatical errors.

Response: We have rewritten this sentence to clarify.

Line 71-73: The connexion between the two sentences is unclear, and the use of the word alternatively is confusing.

Response: We have rewritten both sentences to clarify.

Line 94: What ‘Basecalling’ tool was used? Any option choices?

Response: Basecalling via MinKnow is done by Guppy. There are not options customizable. We have complemented these information.

Figure 3 is presented before figure 4

Response: We have corrected this mistake by reordering.

Line 117: missing comma

Response: Comma added.

Line 124-126: The third objective is a bit confusing. Isn’t the Guppy algorithm already in-use by the nanopore software? What was the intent of the improvement?

Response: That is right, Guppy is already in use. The improvement is that the onboard GPU of the MinION MK1C with it’s designated Cuda cores is actually able to do the basecalling in real time. This isn’t possible with a connected notebook or desktop computer (unless there is an NVIDIA graphics card with Cuda cores, which can be configured to do the basecaling, which needs some expertise). The improving step is selecting the novel Nanopore device with the onboard/faster basecalling abilities.

Line 168-170: Can the authors develop more on this statement? How could a similar surveillance functionality be achieved using RAMPART?

Response: We have complemented: 'RAMPART runs simultaneously with MinKNOW and exhibits demuxing and mapping results in real-time.'

Line 190-191: This is quite concerning. Could the authors develop more on the reasons behind these differences in the phylogenetic analysis based on the bioinformatic software used?

Response: We have added more information and interpretation (from line 251): 'Notably, despite of the calculated different phylogenetic distance using Geneious, the sample was still assigned to clade 20I. We speculate that the reason for this distance bias might be due to the Illumina-optimized assembly pipeline of Geneious.'

Line 227-229: The authors should provide a Ct value threshold for obtaining high-quality consensus sequences. This should also be provided in the abstract as it’s essential information for potential users of the proposed workflow.

Response: We have done this carefully as proposed and described above.

Line 259: The unnecessity to use unspecific hexanucleotide priming is interesting, which is why the authors should develop more on this to give the readers a context of the use

Response: We have added more information in lines 330/331.

Reviewer 3 Report

The Authors described in great detail how to perform SARS CoV-2 sequencing. This method is necessary for the detection of new SASRS CoV-2 variants. It is great importance both from epidemiological , especially novel variants of concern. Moreover, diagnostic CT values are   basic criteria for specimen selection for nanopore sequencing. As descibe the Authors, this method may be used in small hospital laboratories. The Authors described in great detail how to perform SARS CoV-2 sequencing. This method is necessary for the detection of new SASRS CoV-2 variants. It is great importance both from epidemiological , especially novel variants of concern. Moreover, diagnostic CT values are   basic criteria for specimen selection for nanopore sequencing. As descibe the Authors, this method may be used in small hospital laboratories. The Authors described in great detail how to perform SARS CoV-2 sequencing. This method is necessary for the detection of new SASRS CoV-2 variants. It is great importance both from epidemiological , especially novel variants of concern. Moreover, diagnostic CT values are   basic criteria for specimen selection for nanopore sequencing. As descibe the Authors, this method may be used in small hospital laboratories. The Authors described in great detail how to perform SARS CoV-2 sequencing. This method is necessary for the detection of new SASRS CoV-2 variants. It is great importance both from epidemiological , especially novel variants of concern. Moreover, diagnostic CT values are   basic criteria for specimen selection for nanopore sequencing. As descibe the Authors, this method may be used in small hospital laboratories.

Author Response

Dear reviewer 3,

we appreciate very much your fast handling and enjoyed reading your very positive thoughts on our manuscript.